# INFERENCE, PREDICTION, AND ENTROPY RATE OF CONTINUOUS-TIME, DISCRETE-EVENT PROCESSES

## ABSTRACT

The inference of models, prediction of future symbols, and entropy rate estimation of discrete-time, discrete-event processes is well-worn ground. However, many time series are better conceptualized as continuous-time, discrete-event processes. Here, we provide new methods for inferring models, predicting future symbols, and estimating the entropy rate of continuous-time, discrete-event processes. The methods rely on an extension of Bayesian structural inference that takes advantage of neural network's universal approximation power. Based on experiments with simple synthetic data, these new methods seem to be competitive with state-of-the-art methods for prediction and entropy rate estimation as long as the correct model is inferred.

## 1 INTRODUCTION

Much scientific data is dynamic, meaning that we see not a static image of a system but its time evolution. The additional richness of dynamic data should allow us to better understand the system, but we may not know how to process the richer data in a way that will yield new insight into the system in question. For example, we have records of when earthquakes have occurred, but still lack the ability to predict earthquakes well or estimate their intrinsic randomness (Geller, 1997); we know which neurons have spiked when, but lack an understanding of the neural code (Rieke et al., 1999); and finally, we can observe organisms, but have difficulty modeling their behavior (Berman et al., 2016; Cavagna et al., 2014). Such examples are not only continuous-time, but also discrete-event, meaning that the observations belong to a finite set (e.g, neuron spikes or is silent) and are not better-described as a collection of real numbers. These disparate scientific problems are begging for a unified framework for inferring expressive continuous-time, discrete-event models and for using those models to make predictions and, potentially, estimate the intrinsic randomness of the system.

In this paper, we present a step towards such a unified framework that takes advantage of: the inference and the predictive advantages of unifilarity– meaning that the hidden Markov model's underlying state (the so-called "causal state" (Shalizi & Crutchfield, 2001) or "predictive state representation" (Littman & Sutton, 2002)) can be uniquely identified from the past data; and the universal approximation power of neural networks (Hornik, 1991). Indeed, one could view the proposed algorithm for model inference as the continuous-time extension of Bayesian structural inference Strelioff & Crutchfield (2014). We focus on time series that are discrete-event and inherently stochastic.

In particular, we infer the most likely unifilar hidden semi-Markov model (uhsMm) given data using the Bayesian information criterion. This class of models is slightly more powerful than semi-Markov models, in which the future symbol depends only on the prior symbol, but for which the dwell time of the next symbol is drawn from a non-exponential distribution. With unifilar hidden semi-Markov models, the probability of a future symbol depends on arbitrarily long pasts of prior symbols, *and* the dwell time distribution for that symbol is non-exponential. Beyond just model inference, we can use the inferred model and the closed-form expressions in Ref. (Marzen & Crutchfield, 2017) to estimate the process' entropy rate, and we can use the inferred states of the uhsMm to predict future input via a $k$-nearest neighbors approach. We compare the latter two algorithms to reasonable extensions of state-of-the-art algorithms. Our new algorithms appear competitive as long as model inference is in-class, meaning that the true model producing the data is equivalent to one of the models in our search.

In Sec. 3, we introduce the reader to unifilar hidden semi-Markov models. In Sec. 4, we describe our new algorithms for model inference, entropy rate estimation, and time series prediction and test our algorithms on synthetic data that is memoryful. And in Sec. 5, we discuss potential extensions and applications of this research.

## 2 RELATED WORK

There exist many methods for studying discrete-time processes. A classical technique is the autoregressive process, AR-$k$, in which the predicted symbol is a linear combination of previous symbols; a slight modification on this is the generalized linear model (GLM), in which the probability of a symbol is proportional to the exponential of a linear combination of previous symbols (Madsen, 2007). Previous workers have also used the Baum-Welch algorithm (Rabiner & Juang, 1986), Bayesian structural inference (Strelioff & Crutchfield, 2014), or a nonparametric extension of Bayesian structural inference (Pfau et al., 2010) to infer a hidden Markov model or probability distribution over hidden Markov models of the observed process; if the most likely state of the hidden Markov model is correctly inferred, one can use the model's structure to predict the future symbol. More recently, recurrent neural networks and reservoir computers can be trained to recreate the output of any dynamical system through simple linear or logistic regression for reservoir computers (Grigoryeva & Ortega, 2018) or backpropagation through time for recurrent neural networks (Werbos et al., 1990).

When it comes to continuous-time, discrete-event predictors, far less has been done. Most continuous-time data is, in fact, discrete-time data with a high time resolution; as such, one can essentially sample continuous-time, discrete-event data at high resolution and use any of the previously mentioned methods for predicting discrete-time data. Alternatively, one can represent continuous-time, discrete-event data as a list of dwell times and symbols and feed that data into either a recurrent neural network or feedforward neural network. We take a new approach: we infer continuous-time hidden Markov models (Marzen & Crutchfield, 2017) and predict using the model's internal state as useful predictive features.

## 3 BACKGROUND

We are given a sequence of symbols and durations of those symbols, $\ldots, (x_i, \tau_i), \ldots, (x_0, \tau_0^+)$. This constitutes the data, $\mathcal{D}$. For example, seismic time series are of this kind: magnitude and time between earthquakes. The last seen symbol $x_0$ has been seen for a duration $\tau_0^+$. Had we observed the system for a longer amount of time, $\tau_0^+$ may increase. The possible symbols $\{x_i\}_i$ are assumed to belong to a finite set $\mathcal{A}$, while the interevent intervals $\{\tau_i\}_i$ are assumed to belong to $(0, \infty)$. We assume stationarity– that the statistics of $\{(x_i, \tau_i)\}_i$ are unchanging in time.

Above is the description of the observed time series. What now follows is a shortened description of unifilar hidden semi-Markov models, notated $\mathcal{M}$, that could be *generating* such a time series (Marzen & Crutchfield, 2017). The minimal such model that is consistent with the observations is the $\epsilon$-Machine. Underlying a unifilar hidden semi-Markov model is a finite-state machine with states $g$, each equipped with a dwell-time distribution $\phi_g(\tau)$, an emission probability $p(x|g)$, and a function $\epsilon^+(g, x)$ that specifies the next hidden state when given the current hidden state $g$ and the current emission symbol $x$. This model generates a time series as follows: a hidden state $g$ is randomly chosen; a dwell time $\tau$ is chosen according to the dwell-time distribution $\phi_g(\tau)$; an emission symbol is chosen according to the conditional probability $p(x|g)$; and we then observe the chosen $x$ for $\tau$ amount of time. A new hidden state is determined via $\epsilon^+(g, x)$, and we further restrict possible next emissions to be different than the previous emission– a property that makes this model *unifilar*– and the process repeats. See Fig. 1 for illustrations of a unifilar hidden semi-Markov model.

## 4 ALGORITHMS

We investigate three tasks: model inference; calculation of the differential entropy rate; and development of a predictor of future symbols. Our main claim is that restricted attention to a special type of discrete-event, continuous-time model called unifilar hidden semi-Markov models makes all three tasks easier.

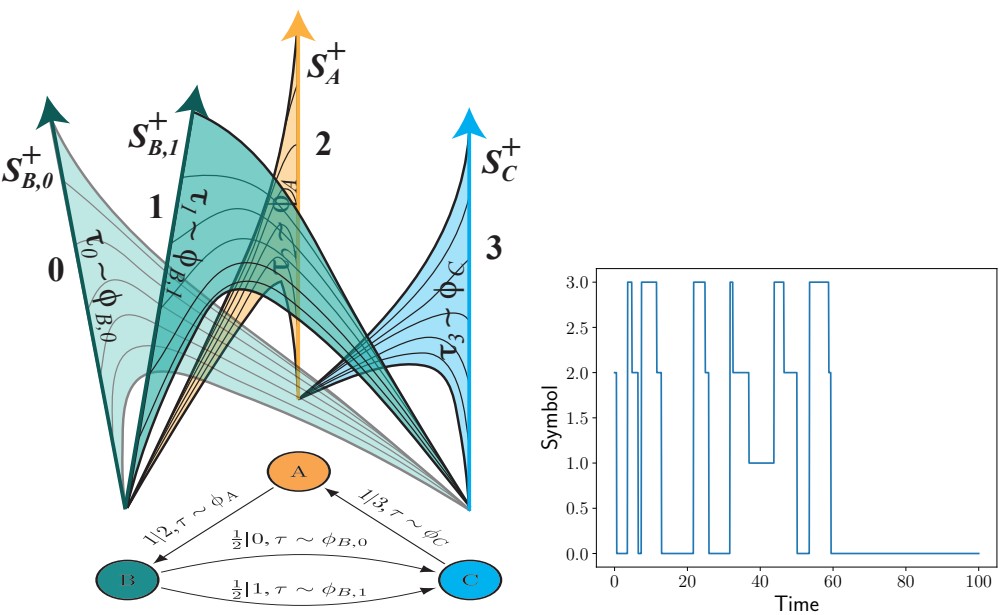

Figure 1: **Illustrations of unifilar hidden semi-Markov models (uhsMm).** At left, two presentations of a uhsMms. At left bottom, a generative model for a discrete-alphabet, continuous-time stochastic process. Dwell times $\tau$ are drawn upon transitions between states, and the corresponding symbol is shown for that amount of time. At top, the corresponding "conveyer belt" representation of the process generated by the model beneath. Conveyer belts represent the time since last symbol based on the height along the conveyer belt traveled; each conveyer belt has a symbol. To the right of the two presentations of a uhsMm, an example time series generated from the model at left, where $\phi_A$, $\phi_B$, $\phi_C$ are inverse Gaussian distributions with $(\mu, \lambda)$ pairs of $(1, 2)$, $(2, 3)$, $(1, 3)$, respectively.

## 4.1 Inference of unifilar hidden semi-Markov processes

The unifilar hidden semi-Markov models described earlier can be parameterized. Let $\mathcal{M}$ refer to a model– in this case, the underlying topology of the finite-state machine and neural networks defining the density of dwell times; let $\theta$ refer to the model's parameters, i.e. the emission probabilities and the parameters of the neural networks; and let $\mathcal{D}$ refer to the data, i.e., the list of emitted symbols and dwell times. Ideally, to choose a model, we would do maximum a posteriori by calculating $\arg\max_{\mathcal{M}} Pr(\mathcal{M}|\mathcal{D})$ and choose parameters of that model via maximum likelihood, $\arg\max_{\theta} Pr(\mathcal{D}|\theta, \mathcal{M})$.

In the case of discrete-time unifilar hidden Markov models, Strelioff and Crutchfield (Strelioff & Crutchfield, 2014) described the Bayesian framework for inferring the best-fit model and parameters. More than that, Ref. (Strelioff & Crutchfield, 2014) calculated the posterior analytically, using the unifilarity property to ease the mathematical burden. Analytic calculations in continuous-time may be possible, but we leave that for a future endeavor. We instead turn to a variety of approximations, still aided by the unifilarity of the inferred models.

The main such approximation is our use of the Bayesian inference criterion (BIC) Bishop (2006). Maximum a posteriori is performed via

$$
\begin{aligned}
BIC &= \max_{\theta} \log Pr(\mathcal{D}|\theta, \mathcal{M}) - \frac{k_{\mathcal{M}}}{2}\log|\mathcal{D}| & (1)\\
\mathcal{M}^* &= \arg\max_{\mathcal{M}} BIC, & (2)
\end{aligned}
$$

where $k_{\mathcal{M}}$ is the number of parameters $\theta$. To choose a model, then, we must calculate not only the parameters $\theta$ that maximize the log likelihood, but the log likelihood itself. We make one further approximation for tractability involving the start state $s_0$, for which

$$
Pr(\mathcal{D}|\theta, \mathcal{M}) = \sum_{s_0} \pi(s_0|\theta, \mathcal{M}) Pr(\mathcal{D}|s_0, \theta, \mathcal{M}). \tag{3}
$$

As the logarithm of a sum has no easy expression, we approximate

$$
\max_{\theta} \log Pr(\mathcal{D}|\theta, \mathcal{M}) = \max_{s_0}\max_{\theta} \log Pr(\mathcal{D}|s_0, \theta, \mathcal{M}). \tag{4}
$$

Our strategy, then, is to choose the parameters $\theta$ that maximize $\max_{s_0} \log Pr(\mathcal{D}|s_0, \theta, \mathcal{M})$ and to choose the model $\mathcal{M}$ that maximizes $\max_{\theta} \log Pr(\mathcal{D}|\theta, \mathcal{M}) - \frac{k_{\mathcal{M}}}{2}\log|\mathcal{D}|$. This constitutes an inference of a model that can explain the observed data.

What remains to be done, therefore, is approximation of $\max_{s_0}\max_{\theta} \log Pr(\mathcal{D}|s_0, \theta, \mathcal{M})$. The parameters $\theta$ of any given model include $p(s', x|s)$, the probability of emitting $x$ when in state $s$ and transitioning to state $s'$, and $\phi_s(t)$, the interevent interval distribution of state $s$. Using the unifilarity of the underlying model, the sequence of $x$'s when combined with the start state $s_0$ translate into a single possible sequence of hidden states $s_i$. As such, one can show that

$$
\begin{aligned}
\log Pr(\mathcal{D}|s_0, \theta, \mathcal{M}) &= \sum_{s}\sum_{j} \log \phi_s(\tau_j^{(s)})\\
&\quad + \sum_{s,x,s'} n(s',x|s)\log p(s',x|s) & (5)
\end{aligned}
$$

where $\tau_j^{(s)}$ is any interevent interval produced when in state $s$. It is relatively easy to analytically maximize with respect to $p(s', x|s)$, including the constraint that $\sum_{s',x} p(s',x|s) = 1$ for any $s$, and find that

$$
p^*(s', x|s) = \frac{n(s', x|s)}{n(s)}. \tag{6}
$$

Now we turn to approximation of the dwell-time distributions, $\phi_s(t)$. The dwell-time distribution can, in theory, be any normalized nonnegative function; inference may seem impossible. However, artificial neural networks can, with enough nodes, represent any continuous function. We therefore represent $\phi_s(t)$ by a relatively shallow (here, three-layer) artificial neural network (ANN) in which nonnegativity and normalization are enforced as follows:

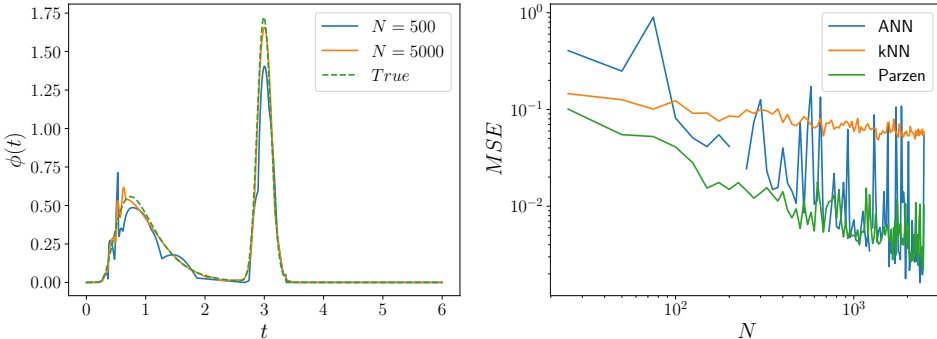

Figure 2: **The estimated density function from varying numbers of samples.** Shown, at left, is the inferred density function using the neural network approach described here compared to the true density function (dotted, green) when given 500 samples (blue) and 5000 samples (orange). As the amount of data increases, the inferred density function becomes closer to ground truth. An interevent interval distribution with two modes was arbitrarily chosen by setting $\phi(\tau)$ to a mixture of two inverse Gaussians. Shown, at right, is the mean-squared error between the estimated density and the true density as we use more training data for three different estimation techniques. The blue line denotes the ANN algorithm introduced here, in which we learn densities from a neural network; the orange line denotes the k-nearest neighbors algorithm (Bishop, 2006; Fukunaga & Hostetler, 1973); and the green line uses Parzen window estimates (Bishop, 2006; Marron et al., 1987). Our new method is competitive with the two standard methods for density estimation.

- the second-to-last layer's activation functions are ReLus ($\max(0, x)$, and so with nonnegative output) and the weights to the last layer are constrained to be nonnegative;

- and the output is the last layer's output divided by a numerical integration of the last layer's output.

The log likelihood $\sum_j \log \phi_s(\tau_j^{(s)})$ determines the cost function for the neural network. Then, the neural network can be trained using typical stochastic optimization methods. (Here, we use Adam Kingma & Ba (2014).) The output of the neural network can successfully estimate the interevent interval density function, given enough samples, within the interval for which there is data. See Fig. 2. Outside this interval, however, the estimated density function is not guaranteed to vanish as $t \to \infty$, and can even grow. Stated differently, the neural networks considered here are good interpolators, but can be bad extrapolators. As such, the density function estimated by the network is taken to be 0 outside the interval for which there is data.

To the best of our knowledge, this is a new approach to density estimation, referred to as ANN here. A previous approach to density estimation using neural networks learned the cumulative distribution function (Magdon-Ismail & Atiya, 1999). Typical approaches to density estimation include $k$-nearest neighbor estimation techniques and Parzen window estimates, both of which need careful tuning of hyperparameters ($k$ or $h$) (Bishop, 2006). They are referred to here as kNN and Parzen. We compare the ANN, kNN, and Parzen approaches in inferring an interevent interval density function that we have chosen, arbitrarily, to be the mixture of inverse Gaussians shown in Fig. 2(left). The $k$ in $k$-nearest neighbor estimation is chosen according to the criterion in Ref. Fukunaga & Hostetler (1973), and $h$ is chosen to as to maximize the pseudolikelihood Marron et al. (1987). Note that as shown in Fig. 2(right), this is not a superior approach to density estimation in terms of minimization of mean-squared error, but it is parametric, so that BIC can be used.

To test our new method for density estimation presented here– that is, by training a properly normalized ANN– we generated a trajectory from the unifilar hidden semi-Markov model shown in Fig. 3(left) and used BIC to infer the correct model. As BIC is a log likelihood minus a penalty for a larger number of parameters, a larger BIC suggests a higher posterior. With very little data, a two-state model shown in Fig. 3 is deemed most likely; but as the amount of data increases, the correct four-state model eventually takes precedence. See Fig. 3(right). The six-state model was never deemed more likely than a two-state or four-state model. Note that although this methodol-

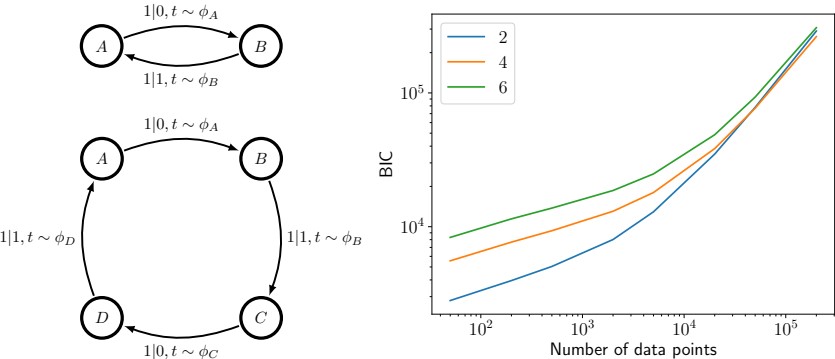

Figure 3: At left, the two-state model (top) and four-state uhsMm (bottom) for binary-alphabet, continuous-time data. At right, BIC as a function of the amount of data for the two-state, four-state, and six-state uhsMms at left. (The six-state uhsMm is not shown.) Larger BIC implies a higher posterior, and so a better fit.

ogy might be extended to nonunifilar hidden semi-Markov models, the unifilarity allowed for easily computable and unique identification of dwell times to states in Eq. 5.

## 4.2 IMPROVED CALCULATION OF DIFFERENTIAL ENTROPY RATES

One benefit of unifilar hidden semi-Markov models is that one can use them to obtain explicit formulae for the differential entropy rate (Marzen & Crutchfield, 2017). Such entropy rates are a measure of the inherent randomness of a process (Crutchfield & Feldman, 2003), and many have tried to find better algorithms for calculating entropy rates of complex processes (Egner et al., 2004; Arnold & Loeliger, 2001; Nemenman et al., 2002; Archer et al., 2014). Setting aside the issue for now of why one would want to estimate the entropy rate, we simply ask how well one can estimate the entropy rate from finite data.

Differential entropy rates are difficult to calculate directly from data, since the usual program involves calculating the entropy of trajectories of some length $T$ and dividing by $T$:

$$h_\mu = \lim_{T \to \infty} \frac{H[\overrightarrow{(x,\tau)}^T]}{T}. \tag{7}$$

A better estimator, though, is the following (Crutchfield & Feldman, 2003):

$$h_\mu = \lim_{T \to \infty} \frac{dH[\overrightarrow{(x,\tau)}^T]}{dT}, \tag{8}$$

or the slope of the graph of $H[\overrightarrow{(x,\tau)}^T]$ vs. $T$. As the entropy of a mixed random variable of unknown dimension, this entropy is seemingly difficult to estimate from data. To calculate $H[\overrightarrow{(x,\tau)}^T]$, we use a trick like that of Ref. Victor (2002) and condition on the number of events $N$:

$$H[\overrightarrow{(x,\tau)}^T] = H[N] + H[\overrightarrow{(x,\tau)}^T|N]. \tag{9}$$

We then further break the entropy into its discrete and continuous components:

$$H[\overrightarrow{(x,\tau)}^T|N=n] = H[x_{0:n}|N=n] + H[\tau_{0:n}|x_{0:n}, N=n] \tag{10}$$

and use the $k$-nearest-neighbor entropy estimator (Kraskov et al., 2004) to estimate $H[\tau_{0:n}|x_{0:n}, N=n]$, with $k$ chosen to be 3. We estimate both $H[x_{0:n}|N=n]$ and $H[N]$ using plug-in entropy estimators, as the state space is relatively well-sampled. We call this estimator model-free, in that we need not infer a model to calculate the estimate.

We introduce a model-based estimator, for which we infer a model and then use the inferred model's differential entropy rate as the differential entropy rate estimate. To calculate the differential entropy

rate from the inferred model, we use a plug-in estimator based on the formula in Ref. (Marzen & Crutchfield, 2017):

$$\hat{h}_\mu = -\sum_s \hat{p}(s) \int_0^\infty \hat{\mu}_s \widehat{\phi}_s(t) \log \widehat{\phi}_s(t) dt, \tag{11}$$

where the sum is over internal states of the model. The parameter $\mu_s$ is merely the mean interevent interval out of state $s$, $\int_0^\infty t\widehat{\phi}_s(t)dt$. We find the distribution over internal states $s$, $\hat{p}(s)$, by solving the linear equations (Marzen & Crutchfield, 2017)

$$p(s) = \sum_{s'} \frac{\mu_{s'}}{\mu_s} \frac{n_{s'\to s}}{n_{s'}} p(s'). \tag{12}$$

We use the MAP estimate of the model as described previously and estimate the interevent interval density functions $\phi_s(t)$ using a Parzen window estimate, with smoothing parameter $h$ chosen so as to maximize the pseudolikelihood (Marron et al., 1987), given that those proved to have lower mean-squared error than the neural network density estimation technique in the previous subsection. In other words, we use neural network density estimation technique to choose the model, but once the model is chosen, we use the Parzen window estimates to estimate the density for purposes of estimating entropy rate. A full mathematical analysis of the bias and variance is beyond the scope of this paper.

Fig. 4 shows a comparison between the model-free method ($k$-nearest neighbor estimator of entropy) and the model-based method (estimation using the inferred model and Eq. 11) as a function of the length of trajectories simulated for the model. In Fig. 4(Top), the most likely (two-state) model is used for the model-based plug-in estimator of Eq. 11, as ascertained by the procedure in the previous subsection; but in Fig. 4(Bottom), the correct four-state model is used for the plug-in estimator. Hence, the estimate in Eq. 11 is based on the *wrong model*, and hence, leads to a systematic overestimate of the entropy rate. When the correct four-state model is used for the plug-in estimator in Fig. 4(Bottom), the model-based estimator has much lower variance than the model-free method.

To efficiently estimate the excess entropy (Crutchfield & Feldman, 2003; Bialek et al., 2001b;a), an additional important informational measure, requires models of the time-reversed process. Future research will elucidate the needed retrodictive representations of unifilar hidden semi-Markov models, which can be determined from the ?forward" unifilar hidden semi-Markov models.

### 4.3 IMPROVED PREDICTORS USING THE INFERRED CAUSAL STATES

There are a wide array of techniques developed for discrete-time prediction, as described earlier in the manuscript. We can develop continuous-time techniques that build on these discrete-time techniques, e.g. by using dwell times and symbols as inputs to a RNN. However, based on the experiments shown here, we seem to gain by first identifying continuous-time causal states.

The first prediction method we call "predictive ANN" or PANN (with risk of confusion with the ANN method for density estimation described earlier) takes, as input, $(x_{-n+1}, \tau_{-n+1}), \ldots, (x_0, \tau_0^+)$ into a feedforward neural network that is relatively shallow (six layers) and somewhat thin (25 nodes). Other network architectures were tried with little improvement. The weights of the network are trained to predict the emitted value $x$ at time $T$ later based on a mean-squared error loss function. For this method to work, the neural network must guess the hidden state $g$ from the observed data, which can be done if the dwell-time distributions of the various states are dissimilar. Increases in $n$ can increase the ability of the neural network to correctly guess its hidden state and thus predict future symbols, assuming enough data to avoid overfitting; here, $n$ is chosen via cross-validation.

The second of these methods that we will label as "RNN" will take $(x_{-n+1}, \tau_{-n+1}), \ldots, (x_0, \tau_0^+)$ as input to an LSTM, though any RNN could have been chosen. The LSTM is asked to produce an estimate of $x$ at time $T$ subject to a mean-squared error loss function, similar to the first prediction method.

The third of these methods that we will label as "uhsMm" preprocesses the input data using an inferred unifilar hidden semi-Markov model so that each time step is associated with a hidden state $g$, a time since last symbol change $\tau_0^+$, and a current emitted symbol $x_0$. In discrete-time applications, there is an explicit formula for the optimal predictor in terms of the $\epsilon$-M; but for continuous-time applications, there is no such formula, and so we use a $k$-nearest neighbor estimate. More precisely,

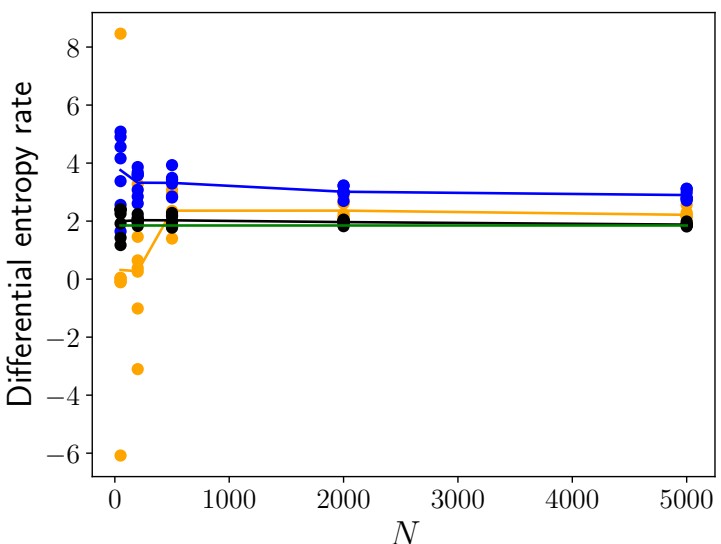

Figure 4: Entropy rate estimation, showing model-free vs. model-based estimators. The synthetic dataset is generated from Fig. 3(top) with $\phi_A(t) = \phi_D(t)$ as inverse Gaussians with mean 1 and scale 5 and with $\phi_B(t) = \phi_C(t)$ as inverse Gaussians with mean 3 and scale 2. The ground truth entropy rate from the formula in (Marzen & Crutchfield, 2017) is 1.85 nats, shown in green. In orange, the model-free estimator (combination of plug-in entropy estimator and kNN (Kraskov et al., 2004) entropy estimators) described in the text. In blue, the model-based estimator assuming a two-state model, i.e., the top left of Fig. 3. In black, the model-based estimator assuming a four-state model, i.e., the bottom left of Fig. 3. Lines denote the mean, and various data point denote the estimated entropy rate for different data sets of a particular size $N$. The model-free method has much higher variance than the model-based methods, and the model-based method for which the correct (four-state) model is used also has lower bias.

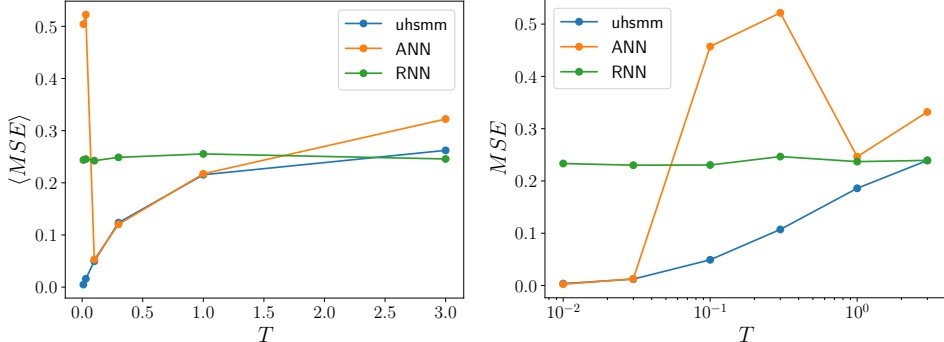

Figure 5: At left, mean-squared error of the predictor of the symbol at time $T$ from data prior to a time of 0 for when 500 data points are available and 300 epochs are used to train the ANN; at right, the mean-squared error of the predictor for when 5000 data points are available and 3000 epochs are used to train the ANN. The generating uhsMm is in Fig. 3(left). The uhsMm method infers the internal state of the unfilar hidden semi-Markov model; the PANN method uses the last $n$ data points $(x_i, \tau_i)$ as input into a feedforward neural network; and the RNN method uses the past $(x_i, \tau_i)$ as input to an LSTM.

we find the $k$ closest data points in the training data to the data point under consideration, and estimate $x_T$ as the average of the future data points in the training set. In the limit of infinite data so that the correct model is identified, for correctly-chosen $k$, this method will output an optimal predictor; we choose $k$ via cross-validation.

The synthetic dataset is generated from Fig. 3(top) with $\phi_A(t) = \phi_D(t)$ as inverse Gaussians with mean 1 and scale 5 and with $\phi_B(t) = \phi_C(t)$ as inverse Gaussians with mean 3 and scale 2. We chose these means and scales so that it would be easier, in principle, for the non-uhsMm methods (i.e., PANN and RNN) to implicitly infer the hidden state ($A$, $B$, $C$, and $D$). Given the difference in dwell time distributions for each of the hidden states, such implicit inference is necessary for accurate predictions. From experiments, shown in Fig. 5, the feedforward neural network and the recurrent neural network are typically outperformed by the uhsMm method. The corresponding mean-squared errors for the three methods are shown in Fig. 3(bottom) for two different dataset sizes. Different network architectures, learning rates, and number of epochs were tried; the results shown in Fig. 5 are typical. Using a k-nearest neighbor estimate on the causal states (i.e., the internal state of the uhsMm) to predict the future symbol requires little hyperparameter tuning and outperforms compute-intensive feedforward and recurrent neural network approaches.

## 5 DISCUSSION

We have introduced a new algorithm for inferring causal states (Shalizi & Crutchfield, 2001) of a continuous-time, discrete-event process using the groundwork of Ref. (Marzen & Crutchfield, 2017). We have introduced a new estimator of entropy rate that uses the causal states. And finally, we have shown that a predictor based on causal states is more accurate and less compute-heavy than other predictors.

The new inference, estimation, and prediction algorithms could be used to infer a predictive model of complex continuous-time, discrete-event processes, such as animal behavior, and calculate estimates of the intrinsic randomness of such complex processes. Future research could delve into improving estimators of other time series information measures (James et al., 2011), using something more accurate than BIC to calculate MAP models, or enumerating the topology of all possible uhsMm models for non-binary alphabets (Johnson et al.).

ACKNOWLEDGMENTS

This material is based upon work supported by, or in part by, the U. S. Army Research Laboratory and the U. S. Army Research Office under contracts W911NF-13-1-0390 and W911NF-18-1-0028 and the Moore Foundation.

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
