# OpenReview forum: "INFERENCE, PREDICTION, AND ENTROPY RATE OF CONTINUOUS-TIME, DISCRETE-EVENT PROCESSES"
_ICLR.cc/2020/Conference — Reject_

### Official Review · AnonReviewer2 · 2019-10-22
**Official Blind Review #2**

**Rating:** 1

**Review:**

The authors present a model for time series which are represented as discrete events in continuous time and describe methods for doing parameter inference, future event prediction and entropy rate estimation for such processes. Their model is based on models for Bayesian Structure prediction where they add the temporal dimension in a rigorous way while solving several technical challenges in the interim. The writing style is lucid and the illustrations are helpful and high quality.

However, the paper and the technical challenges which the authors faced have been extensively studied in the closely related field of (marked) temporal point process modeling, which also models processes with discrete events in continuous time. Hence, the claim "When it comes to continuous time, discrete-event predictors, far less has been done" is mildly erroneous. The literature in that field has provided several ways of handling the technical challenges which authors have tried to overcome. For example, excellent predictor for the next event as well as time has been proposed by Du. et. al. (2016), Mei and Eisner (2017) and, very recently, by Türkmen et. al (2019), among many many others. Also, probability model (i.e. \phi_s(t)) of the next event time can be represented using intensity functions \lambda(t). However, if the authors require an explicit likelihood function, Normalized Flows with re-parametrization, can provide that too (see Kobyzev (2019) for a review of recent work in the area). This while being a Neural Network based approach well-weathered models and does not require numerical integration. Without placing this work in context of these works, it is very difficult to judge the contributions of the paper well.

Then there are a few unanswered questions about the model:

 1. In the Background, where the authors describe the generative process, does the emitted symbol also depend on the dwell time \tau (like Mei and Eisner, 2017), or is the distribution of the dwell time and the next symbol independent given the state (like Du et. al)?
 2. Similarly, during entropy calculation, while breaking the entropy term from (9) to (10), shouldn't there be a dependence of the discrete symbols on the continuous dwell times of the prior step?
 3. We need significantly more information about the training used for the LSTM/RNN models to be able to judge whether the performance of the methods is comparable or not. These could be provided in a supplementary material, if they do not fit within the main text.


Some other ways of improving the paper:
 - The reference style changes frequently in the paper and is mutually inconsistent.
 - The \pi in Eqn. (3) is not formally defined. If it is the initial state probability, then should it be dependent on the model \Mcal and parameters \theta?
 - The illustrations, though useful, are too large and use up space.
 - Though the task of determining the entropy is indeed intellectually satisfying, it would very much help motivate the reader if there were some applications which the authors could allude to.


Citations:
 - Aalen, Odd, Ornulf Borgan, and Hakon Gjessing. Survival and event history analysis: a process point of view. Springer Science & Business Media, 2008.
 - Du, Nan, et al. "Recurrent marked temporal point processes: Embedding event history to vector." Proceedings of the 22nd ACM SIGKDD International Conference on Knowledge Discovery and Data Mining. ACM, 2016.
 - Mei, Hongyuan, and Jason M. Eisner. "The neural hawkes process: A neurally self-modulating multivariate point process." Advances in Neural Information Processing Systems. 2017.
 - Türkmen, Ali Caner, Yuyang Wang, and Alexander J. Smola. "FastPoint: Scalable Deep Point Processes."
 - Kobyzev, Ivan, Simon Prince, and Marcus A. Brubaker. "Normalizing flows: Introduction and ideas." arXiv preprint arXiv:1908.09257 (2019).


**Experience Assessment:**

I have published in this field for several years.

**Review Assessment: Checking Correctness Of Derivations And Theory:**

I assessed the sensibility of the derivations and theory.

**Review Assessment: Checking Correctness Of Experiments:**

I assessed the sensibility of the experiments.

**Review Assessment: Thoroughness In Paper Reading:**

I read the paper at least twice and used my best judgement in assessing the paper.

---

### Official Review · AnonReviewer3 · 2019-10-23
**Official Blind Review #3**

**Rating:** 3

**Review:**

This paper proposed a model for continuous-time, discrete events prediction and entropy rate estimation by combining unifilar hidden semi-Markov model and neural networks where the dwell time distribution is represented by a shallow neural network.

Comments:
The literature review on previous work for continuous-time, discrete events prediction is not thorough enough. For this problem, there are continuous-time Markov networks [El-Hay et al. 2006], continuous-time Bayesian networks [Nodelman et al. 2002] and its counterpart in relational learning domain, i.e. relational continuous-time Bayesian networks [Yang et al. 2016]. The authors should have learned their work and addressed the difference between the proposed model and these work in the related work section.

Tal El-Hay, Nir Friedman, Daphne Koller, and Raz Kupferman. Continuous Time Markov Networks. In UAI, 2006.
Nodelman, U.; Shelton, C.; and Koller, D. Continuous Time Bayesian Networks. In Proceedings of the Eighteenth Conference on Uncertainty in Artificial Intelligence (UAI), pages 378–387, 2002.
Shuo Yang, Tushar Khot, Kristian Kersting, and Sriraam Natarajan. Learning continuous-time Bayesian networks in relational domains: A non-parametric approach. In Proceedings of the Thirtieth AAAI Conference on Artificial Intelligence, pages 2265–2271, 2016.

The approximations made to get equation (3) and equation (4) lacks theoretical proof for the up-bound of its influence on the model performance. If these assumptions have been made before, please cite the references; if not, please illustrate why it makes sense and how the approximation will influence the value of the objective function.

Please explicitly state the meaning of all the symbols used in the paper even it can be inferred by the readers. E.g. ‘n’, which first appears in Equation (5) and is never being explained.

The experiments are rather simple both in terms of the model used to generate the data and the number of different data sets being used. Hence, the experimental results are not strong enough to support the claims made by this paper.
Specifically, it claims “With very little data, a two-state model shown in Fig. 3 is deemed most likely; but as the amount of data increases, the correct four-state model eventually takes precedence”, but in Figure 3, the plot with the highest BIC score when the training sample is less is the green curve which stands for the six-state model according to the legend.
“The corresponding mean-squared errors for the three methods are shown in Fig. 3(bottom) for two different dataset sizes.” I could not find it in Figure 3.


**Experience Assessment:**

I have published one or two papers in this area.

**Review Assessment: Checking Correctness Of Derivations And Theory:**

I carefully checked the derivations and theory.

**Review Assessment: Checking Correctness Of Experiments:**

I carefully checked the experiments.

**Review Assessment: Thoroughness In Paper Reading:**

I read the paper at least twice and used my best judgement in assessing the paper.

---

### Official Review · AnonReviewer1 · 2019-10-28
**Official Blind Review #1**

**Rating:** 1

**Review:**

The paper focuses on the problem of modeling, predicting and estimating entropy information over continuous-time discrete event processes. Specifically, the paper leverages unifilar HSMM's for model inference and then uses the inferred states to make future predictions. The authors also use inferred model with previously developed techniques for estimating entropy rate. The authors describe the methods and provide the evidence of the effectiveness of their method with experiments on a synthetic dataset.

The paper should be rejected as it doesn't appear to be fit for this conference. While the paper does build techniques that leverages classical statistical learning and combine them with deep learning, it falls short on various aspects. Specifically, the paper neither provides any rigorous analysis of the developed methods so as to be useful to theoretical learning community nor does it support its approaches with strong empirical evidence so as to be of practical importance.

- While the authors cite lack of work in such data setting (which is actually incorrect as described below), they do not motivate or justify why such an approach of using either unifilar HSMM or Bayesian inference techniques are good starting point to model such information. The authors claim that unifilar HSMM is more of a restriction to make the three proposed tasks easier. But such a model assumption is very strict and will likely lead to large model mis-specification issues when used in real-world. The author does not describe what can be done in this scenario.

- Page2, Section 3: the statistics of {(x_i, t_i)} are unchanging in time. What does this mean?
- As stated by authors, to make the approach tractable, they have to adopt a slew of approximations which seems to put severe restriction on what kind of data can be modeled and inferred on using these approaches.
- It is not clear why one wants to estimate the entropy rate in such settings and authors avoid discussing it in Section 4.2
similarly the authors explicitly avoid discussing bias-variance tradeoff of such method, however, this makes the paper
devoid of any insights into the proposed approach.
- How was k chosen to be 3 in Eq 10?
- Another major miss for the paper is that there has been lot of work at the intersection of stochastic processes (e.g. temporal point processes) and machine learning techniques that aim to model exactly the continuous-time discrete event information and the paper does neither cites any of them nor discusses them or compare with them. We encourage the authors to look at  the relevant literature and position their work in comparison to these existing approaches. I provide a few references here as starting point, however, lot of follow-up work are available:

Point Processes and Machine Learning:
------------------------------------
Modeling the Dynamics of Learning Activity on the Web
Smart Broadcasting: Do You Want to be Seen?, Karimi et. al.
Uncovering the Temporal Dynamics of Diffusion Networks, Gomez-Rodriguez et. al.

Hidden semi-markov models:
-------------------------
Recurrent Hidden Semi-Markov Model, Dai et. al.

Deep learning and temporal modeling:
-----------------------------------
Recurrent Marked Temporal Point Processes: Embedding Event History to Vector, Du et. al.
The Neural Hawkes Process: A Neurally Self-Modulating Multivariate Point Process, Mei et. al.

- Finally, the authors show only one synthetic experiment. This is far from providing convincing evidence of the efficacy of the method and certainly needs a lot more work for acceptance in any machine learning conference.  The authors, at the least, must use their approaches on couple of real-world dataset (e.g. earthquake information, animal behavior as they mention or others) and also compare with the above stochastic processes techniques to test how their methods fare.

- Also, the analysis of efficiency of their method is required. It appears that their methods may not be scalable for instance to a multi-dimensional discrete label case.

- Figure 4 is not correctly referenced in the paragraph right above Section 4.3.

**Experience Assessment:**

I have published in this field for several years.

**Review Assessment: Checking Correctness Of Derivations And Theory:**

I assessed the sensibility of the derivations and theory.

**Review Assessment: Checking Correctness Of Experiments:**

I carefully checked the experiments.

**Review Assessment: Thoroughness In Paper Reading:**

N/A

---

### Decision · Program_Chairs · 2019-12-19

**Decision:**

Reject

**Comment:**

The authors present a Bayesian model for time series which are represented as discrete events in continuous time and describe methods for doing parameter inference, future event prediction and entropy rate estimation for such processes.
However, the reviewers find that the novelty of the paper is not high enough, and without sufficient acknowledgement and comparison to existing literature.